# Lipid Characteristics of the Muscle and Perirenal Fat in Young Tudanca Bulls Fed on Different Levels of Grass Silage

**DOI:** 10.3390/ani13020261

**Published:** 2023-01-12

**Authors:** Javier Mateo, Irma Caro, Sergio Soto, Sonia Andrés, Rocío Aláiz-Rodríguez, Emma Serrano

**Affiliations:** 1Departamento deHigiene y Tecnología de los Alimentos, Facultad de Veterinaria, Universidad de León, Campus Vegazana s/n, 24007 León, Spain; 2Facultad de Medicina, Universidad de Valladolid, 47005 Valladolid, Spain; 3Area Academica de Medicina Veterinaria y Zootecnia, Instituto de Ciencias Agropecuarias, Universidad Autónoma del Estado de Hidalgo, Ave. Universidad s/n km 1, Ex-Hacienda de Aquetzalpa, Tulancingo 43600, Hidalgo, Mexico; 4Instituto de Ganadería de Montaña, CSIC-Universidad de León, Finca Marzanas s/n, 24346 Grulleros, Spain; 5Departamento de Ingeniería Eléctrica y de Sistemas y Automática, Escuela de Ingenierías Industrial, Informática y Aeroespacial, Universidad de León, Campus Vegazana s/n, 24007 León, Spain; 6Dirección General de Desarrollo Rural, Centro de Investigación y Formación Agrarias, Gobierno de Cantabria, C/Héroes Dos de Mayo, 27, 39600 Muriedas, Spain

**Keywords:** authentication, beef, fat-soluble vitamins, fatty acids, feeding system, volatile compounds

## Abstract

**Simple Summary:**

Grass silage is commonly used as a forage source in winter for finishing bull in European regions under oceanic climates, such as the Cantabrian cornice, where the local Tudanca breed is produced. Grass silage, compared with barley straw in the bull’s diet showed an effect on beef quality, i.e., increased saturated fatty acids, n-3 fatty acids, and β-carotene and decreased monounsaturated and polyunsaturated fatty acids. Beef from silage-fed bulls and straw-fed bulls could be differentiated by fatty acid percentages, especially 18:0, *t*-18:1, and *c9*-18:1, b-carotene content, b* colour value, and carotenoid colour index.

**Abstract:**

Beef derived from grass-fed cattle is a specific quality criterion. The effect of grass silage intake on quality characteristics, i.e., fatty acids, fat-soluble vitamins, and lipid-derived volatile composition of intramuscular and perirenal fat from fattening bull weaners were studied. Visible (VIS) and near-infrared (NIR) spectra were also obtained from perirenal fat. Perirenal fat analysis was performed for feeding differentiation purposes. A total of 22 Tudanca breed 11-month-aged bulls were finished on three different diets: grass silage and a commercial concentrate ad libitum (GS-AC), grass silage ad libitum and the commercial concentrate restricted to half of the intake of the GS-AC group (GS-LC), and barley straw and concentrate ad libitum (Str-AC). Feeding had a significant effect (*p* < 0.05) on γ-linolenic acid and the ratio n-6/n-3 fatty acids. Furthermore, β-carotene content was greater in beef from silage groups than in the Str-AC group. Feeding also affected the perirenal fat composition. Beef from silage-fed bulls and straw-fed bulls could be differentiated by fatty acid percentages, especially 18:0, *t*-18:1, and *c9*-18:1, β-carotene content, b* colour value, and carotenoid colour index. However, the VIS or NIR spectra data showed poor differentiating performance, and the volatile composition did not have appreciable differentiation power.

## 1. Introduction

Bulls can be finished using either grain-based or forage-based diets. Moreover, different sources of forage can be used, such as straw, grass silage, and fresh grass. The feed composition in finishing bulls’ diets has an influence on beef quality [1,2]. Moreover, a high amount of grass in cattle diets elicits positive perceptions regarding animal welfare, environmental impact, nutritional value, and meat quality in producers and consumers [3]. Beef from cattle fed on high amounts of grass can be more appreciated than conventional beef in specific niche markets, where the high intake of grass by bulls can be a beef quality criterion [4,5]. From an animal welfare perspective, a large enough quantity of forage, i.e., diets containing more than 25% of neutral detergent fiber, is necessary to avoid metabolic disorders present when conventional concentrated-based diets are used [6]. Regarding beef quality, several studies have shown that the amount of grass intake can affect the fatty acid (FA) composition and natural antioxidant presence in beef, and thus affect the beef flavour and oxidative stability [7,8,9]. The magnitude of this effect, however, would depend on not only the grass intake but also the grass type.

Authentication methods to guarantee the grassland origin of beef have received recent research attention. In this context, a number of studies have explored the performance of different analytical tools in beef authentication regarding specific cattle feeding systems, i.e., dietary intakes of grass, grass silage, or concentrate [10,11,12,13,14,15,16]. Stable isotopes, FAs, lipid-soluble vitamins, or volatile compounds have been the analytes used for differentiation purposes. Although these approaches have proven to be useful and effective to some extent, one main difficulty seems to be to cope with the variations in quantity, physical characteristics, and composition of forage feedstuffs consumed by cattle [11]. While most of the above-mentioned studies observed fresh-grass feeding systems, few considered grass-silage feeding systems, which suggests the need for further research on the latter systems.

Grass silage is a predominant forage source for cattle in European regions under oceanic climates, such as the Cantabria cornice [17]. The availability of grazing surfaces in this region’s farms can be limited in winter, when animals need to be indoors due to bad weather conditions and low natural feeding resources. In a previous study [18], with the current study being a continuation of it, the effects of three different silage rates, i.e., no silage, medium, and high, in the finishing diets of Tudanca (a Cantabrian local breed) bull weaners on growth, carcass characteristics, economic performance, and beef quality was studied. The results obtained suggested that the weaner bulls should be fed the high grass-silage diets in indoor systems during winter. As an extension of the previous source [18], this study presents two additional aims. The first aim was to better understand the effect of the level of grass silage in the diet of weaner bulls on beef quality by focusing on intramuscular lipid composition and lipid-derived volatile compounds in the headspace of cooked meat. The second aim was to investigate the potential of the compositional and spectral characteristics of perirenal fat for differentiation between the three feeding systems with different levels of silage.

## 2. Materials and Methods

### 2.1. Animal Material

The meat and fat sampled from 22 Tudanca bull carcasses were used in this study. The rearing conditions of the bulls whose carcasses were sampled were those previously described by Serrano et al. [18]. In brief, 22 calves were raised on pasture with their mothers until weaning (5 months of age). The animals were randomly assigned to three groups, each placed in a feedlot pen, and fed on hay and commercial concentrate ad libitum for a 15-day adaptation period. Each group was then assigned to one of the following three diets: grass silage and commercial concentrate ad libitum (GS-AC; *n* = 7), grass silage ad libitum, and commercial concentrate limited to half of the intake of the GS-AC group (GS-LC, *n* = 7), and barley straw and commercial concentrate ad libitum as a control group (Str-AC, *n* = 8). Grass silage consisted of 2nd year-spring *Lolium perenne* growths wilted to a dry matter content of 30% and cut to 12 cm length. The animals were reared in individual pens on these diets for 143 ± 4 days until slaughter at 11 months of age, a usual age for slaughtering in Spain [19]. The mean (standard deviation) body weights of the bull weaners at slaughtering were respectively 299 ± 12, 253 ± 15, and 285 ± 11 for GS-AC, GS-LC, and Str-AC. The proximate composition, net energy content, and fatty acid percentages of the concentrate, grass silage, and barley straw were described in Serrano et al. [18]. For the analysis of fat-soluble vitamins, the feeds were sampled on two different days (one at the beginning and the other by the middle of the experimental period; approximately 200 g of concentrate, 100 g of silage, and 25 g of a straw). Samples were frozen, lyophilised, and finely powdered with a knife mill, and the concentrations of the fat-soluble vitamins were determined following the methods explained below. Table 1 shows the ether extract, FA percentages, and vitamin contents of feeds. The mean daily intake of the feeds, expressed in kg of dry matter, were the following: 4.3 concentrate and 1.4 grass silage for GS-AC, 2.6 concentrate and 2.1 grass silage for GS-LC, and 4.9 concentrate and 0.5 barley straw for Str-AC.

At 24 h post-mortem, two 2-cm steaks from the *longissimus thoracis* (LT), one at the 6th and the other at the 7th ribs, and the whole perirenal fat depot were sampled from the right-hand half carcasses. The samples were transported into isothermal boxes to the lab in the Centro de Investigación y Formación Agrarias (CIFA; Muriedas, Cantabria, Spain). There, the steaks were packaged under a vacuum, stored (4 °C) for 7 days for ageing, and then frozen (−32 °C). The perirenal fat portions were individually vacuum packaged and frozen (−32 °C). The LT steaks and perirenal fat portions were isothermally transported to the University of León, where they were kept frozen until further analysis.

Before analysis, the LT steaks were thawed at 4 °C for 48 h. One steak was homogenised with a domestic blender, and two 1-g aliquots were lyophilised and then used for FA analysis. The rest was used for the analysis of fat-soluble vitamins. The other steak was cooked in a pre-warmed (165 °C) clam-shell grill to a core temperature of 75 °C (measured by a Digi-Sense thermocouple probe, Cole-Parmer Instrument Company, Vernon Hills, IL, USA) and then divided into two equal portions. One portion was used for the immediate analysis of volatiles and thiobarbituric acid reactive substances (TBARS) in recently cooked meat (Day 0), and the other was stored in the dark at 4 °C ± 2 °C covered with polyvinylchloride cling film for two days and then analysed for TBARS (Day 2).

A 50 g sample of perirenal fat, after being thawed (24 h, 4 °C), was blended with a domestic blender, and aliquots were taken for the determination of FAs, lipid-soluble vitamins, and volatile composition. The remainder were analysed for instrumental colour and near-infrared reflectance (NIR) spectroscopy measurements.

### 2.2. Chemical Analysis

FAs in muscle and perirenal fat samples (250 mg of lyophilised meat and 50 mg of perirenal fat, respectively) were analysed in duplicate following the procedure described by Andrés et al. [20], with a modification in the helium flow, which was 3 mL/min; data were first calculated as a percentage of FAs of the total FAs and then percentages were transformed in mg/100 g of meat or perirenal fat using the fat content of the samples and the 0.95 (intramuscular fat) and 0.92 (perirenal fat) lipid conversion factors [21]. Volatile compounds in the headspace of cooked muscle and raw perirenal fat samples were determined in duplicate. Samples (5 g) were placed in 20 mL headspace vials sealed with magnetic screw silicone/PTFE septa caps (Agilent Technologies, Santa Clara, CA, USA). The extraction procedure (direct static headspace), chromatographic separation, detection, and quantification followed methods previously described by Vieira et al. [22]. Concentrations of volatiles were expressed as ng equivalents of undecane per ml of headspace. The oxidative stability of meat lipids was assessed by TBARS analysis [23] of recently cooked and two-day refrigerated stored cooked meat. The results were expressed as due to the storage increase (ΔTBARS) in mg of malondialdehyde per kg of meat.

The fat-soluble vitamins: β-carotene, retinol, and tocopherols, were analysed in animal feeds (silage, concentrate, and straw), meat, and perirenal fat (only β-carotene and retinol) using high-performance liquid chromatography. For the analysis of β-carotene and retinol, 1-g samples of minced meat and 0.5-g samples of perirenal fat (and 0.5 g of homogenised animal feeds) were subjected to saponification and extraction following the procedure described by Mestre-Prates et al. [24]. The filtered hexane solution was then subjected to chromatographic analysis, according to Jin et al. [25]. Tocopherols were extracted (from 1 g of meat and 0.5 g of feed) and analysed using the method described by Humada et al. [26], with the difference that the external standard method was used for quantification.

The fat content of the perirenal fat depot was determined in duplicate according to the AOAC ([27] (Official method no. 991.36)) with slight modifications. A sample of 5.0 g of minced fat was mixed with 25.0 g of sea sand and 10 mL of 96% ethanol in stainless steel crucibles. After the ethanol was evaporated using a sea bath at 90 °C, the mixture was dried for 4 h at 90 °C and, finally, fat was extracted with petroleum ether. The colour of the perirenal fat was analysed in triplicate on the cut surface of a two-cm-thick fat slice obtained with a knife from the fat depot, 1 h after cutting and avoiding blood blots. Colour was determined using a CM-700d (Konica Minolta Sensing Inc., Osaka, Japan) apparatus operating with a D65 illuminant, SCI mode, 11 mm aperture for illumination, and 8 mm for measurement, and 10° visual angle. The colorimetric variables of the CIE L*a*b* uniform colour space and the reflectance spectra, i.e., the visible (VIS) spectral data, were recorded. From the reflectance spectra, spectral data between 450 and 510 nm were used to calculate the I450-510 index proposed by Prache et al. [3] to quantify the light absorption by carotenoid pigments in solid animal fat.

Fat samples were also scanned at 2-nm intervals over the near-infrared (NIR) spectral range (1100–2500 nm) using an InfraAlyzer 500-model spectrophotometer (Bran+Luebbe GmbH, Norderstedt, Germany). A 0.5 g sample at 30 °C was placed in a golden cup (Bran+Luebbe) and covered with a slide glass. Each sample was measured twice. The signal coming from the instrument was converted to absorbance (A = log 1/transmittance), and the two measurements obtained were averaged for each sample. The instrument was operated using SESAME software version 1 (Bran+Luebbe GmbH, Norderstedt, Germany). The measurement conditions and the treatment of spectral data, i.e., elimination of outliers and scatter correction, were carried out as described in Prieto et al. [28].

### 2.3. Statistical Analysis

Meat and fat characteristics for the different feeding treatments were statistically compared using a one-way analysis of variance where diet was used as a fixed factor. When the *p*-value was lower than 0.05 (significance level), the least square difference test was carried out to analyse the significance of the paired comparisons (*p* < 0.05). Moreover, a Pearson correlation analysis was carried out between the intramuscular fat content of the samples [18] and their FA percentages. Analyses were performed with the SPSS 17.0 (IBM Corp., Armonk, NY, USA) package.

VIS and NIR reflectance spectral data obtained from each fat sample were analysed using MATLAB and StatisticToolbox (released 2018b). Principal component (PC) analysis was carried out using the major FAs, i.e., those with percentages higher than 0.4% of total FAs. Another PC analysis was carried out for the VIS and NIR absorbance spectra, considered individually or jointly, and this was also done with their first derivatives with respect to wavelength. In the spectra PC models, all wavelengths were considered variables. Analyses were performed by centering the data using the singular value decomposition (SVD) algorithm [29].

## 3. Results and Discussion

### 3.1. Intramuscular Fat Composition

The FA profile of the *longissimus thoracis* muscle is shown in Table 2. Overall, no significant differences (*p* > 0.05) in the level of individual FAs or their sums were detected between GS-AC and GS-LC beef. Compared to straw (Str-AC), feeding grass silage (either GS-AC or GS-LC) resulted in significantly reduced levels of γ-linoleic acid (18:2n − 6; *p* = 0.03 or *p* = 0.01, respectively) as well as the sum of *t10*-18:1 and vaccenic acid (*t11*-18:1; *p* = 0.02 or *p* = 0.009), adrenic acid (22:4n − 6; *p* = 0.007 or *p* = 0.01), and n-6/n-3 ratio (*p* < 0.001).

To explain the FA results, it should be considered that the variations on the FA profile of beef intramuscular fat present multiple causalities, i.e., dietary FA intake, rumen fermentation, and microbial biohydrogenation of ingested PUFA [30,31], and amount of intramuscular fat, which increases the proportion of neutral lipids with regards to phospholipids in muscle [32]. Regarding the variations in FA intake due to grass (or grass silage)/concentrate ratio in the diet, different studies [7,12,32,33] have reported increased grass intakes (or reduced concentrate intakes) to increase n − 3 FA levels in the intramuscular fat of beef. In contrast, in this study, n − 3 FA levels were not affected by the amount of dietary grass silage (*p* = 0.841). This is in spite of the fact that the dietary intake of 18:3n – 3 was approximately 20% higher for the animals from GS-AC and GS-LC groups than from those of the straw-fed animals [18]. However, the greater intake of 18:3n − 3 by the grass silage-fed animals could be responsible, at least partially, for the reduced n − 6/n − 3 ratios in the beef from these animals.

However, despite the differences in dietary PUFA and 18:3n − 3 intakes, no significant differences were found either for PUFA or n − 3 FA contents in the beef between any treatments. The overlapping effects of other factors, such as intramuscular fat content, could explain the lack of effect of dietary PUFA levels on beef PUFA levels. Intramuscular fat content has a key effect on the beef FA profile, i.e., more intramuscular fat is related to greater FA levels, but reduced PUFA percentages, and those greater SFA and MUFA percentages [32]. In this study, the intramuscular fat content of beef was 2.5%, 2.3%, and 2.7% for, respectively, GS-AC, GS-LC, and Str-AC [18]. Carrying on a Pearson correlation analysis to the 22 individual samples, intramuscular fat content explained almost 100% of SFA and MUFA and 43% of PUFA levels (mg/100 g of beef), i.e., the r (Pearson) coefficients between intramuscular fat content and SFA, MUFA, and PUFA levels were 0.988, 0.994, and 0.660, respectively. Moreover, intramuscular fat content explained approximately 30% of the variation in PUFA percentage among individual samples, i.e., the r (Pearson) coefficients between intramuscular fat content and PUFA, n − 3 and n − 6 FA percentages were −0.55 (*p* < 0.05), −0.62 (*p* < 0.05), and −0.52 (*p* < 0.05), respectively.

It is generally accepted that a positive relationship between grass (or grass silage) feeding and the levels of both conjugated linoleic fatty acids (CLA) and *t*-18:1 FAs in beef [7,12,33]. However, the presence of these two biohydrogenation intermediates depends not only on the amount of forage in the diet but also on the type and amount of forage, the total PUFA intake, and the starch content in the diet [34]. Thus, the positive relationship between grass vs CLA and *t*-18:1 C has not always been evidenced. Warren et al. [32] found that the greater grass silage in steers’ diets, the reduced the levels of both *t*-18:1 and CLA. These authors resorted to a set of possible reasons for explaining the contrasting results with previous studies, among which the more relevant is the differential effect of fresh grass from grass silage on microbial biohydrogenation. Fresh grass, rich in readily fermentable sugars, and soluble fibre compared to grass silage, provides specific ruminal conditions promoting the biohydrogenation of PUFA, and hence the formation of *t*-18:1 and CLA. Similarly, in this study, CLA levels were not affected by the amount of grass silage intake (the levels were similar between GS-AC and GS-LC). Furthermore, the Str-AC group showed the highest *t*-18:1 percentage. Several factors could account for this finding, such as the increased PUFA intake in Str-AC and GS-LC and rumen microbial activity, which might have been more efficient in producing *t*-18:1 FAs.

The fat-soluble vitamin content in the muscle is also shown in Table 2. Only the silage-fed animals presented a detectable β-carotene content, which is due to the high content of this provitamin in the grass silage and in fresh grass. Greater levels (i.e., 4–16-fold) of β-carotene in beef from grass-fed cattle compared to beef from cereal-fed cattle have been reported in different studies [7]. Despite the differences in β-carotene, the levels of retinol, which in part are formed in animal tissues from β-carotene, did not show significant differences. No differences in α-tocopherol contents were found between the experimental treatments, and δ- and γ-tocopherol (not shown in the table) were under the detection limit (0.1 µg/g). Both silage and concentrate were important sources of α-tocopherol (around 40 µg/g dry matter, Table 1). These high levels in the concentrate are explained by the use of α-tocopheryl acetate as an additive.

Feeding system effects on FA and fat-soluble vitamins affect the nutritional value of intramuscular lipids. On the one hand, the most favourable PUFA/SFA ratio was found for the Str-AC group (0.43; the nearest to the recommended value of 1 for the diet as a whole; [35], followed by GS-LC (0.36), and finally GS-AC (0.26). On the other hand, the most favourable n − 6/n − 3 ratio was for beef from grass silage-fed bulls (8.4, GS-AC, and 7.5, GS-LC) compared to that from straw-fed bulls (13.1, Str-AC), with all of them over the recommended ratio 1-2/1 for a healthy diet [35]. In addition, grass-silage beef had higher amounts of CLA, associated with positive effects on health [34]. The presence of higher amounts of *t*-18:1 in Str-AC beef would be inconclusive, as their effect on health depends on the specific *t*-FA [34]. Furthermore, *t*11-18:1 is considered a positive FA precursor of CLA in humans; however, *t10*-18:1, which can be more abundant than *t11*-18:1 in grain-based diets, appears to have a negative impact on health [7,34]. Silage feeding presented a favourably greater β-carotene content, which is a precursor of retinol. Overall, among the three groups, GS-LC beef was considered the selected option regarding nutritional quality; despite the intermediate value in the PUFA/SFA ratio, it showed a relatively high n − 6/n − 3 ratio, CLA, and β-carotene.

TBARS increments (ΔTBARS) during refrigerated storage of cooked meat, also shown in Table 2, was not significantly different between feeding groups. The oxidation of cooked beef samples during refrigerated storage would not depend on the differences found in PUFA content (Table 2), which are the FAs most susceptible to lipid oxidation reactions. A total of 45 volatile compounds belonging to the following chemical families: aldehydes (13), alcohols (6), ketones (4), aliphatic hydrocarbons (7), furans (1), FAs and lactones (2), benzene compounds, terpene compounds (5), and sulphur compounds (3), were detected in the cooked beef headspace. For brevity, Table 3 shows those compounds with levels higher than 0.5 ng equivalents of undecane/mL of headspace. Most of the compounds, i.e., the aliphatic aldehydes, pentanol, hexanol, heptanol, 1-octen-3-ol and octanol, heptanone, methyl-heptanone, octanedione and penthylfuran, presumably originated by thermal lipid oxidation [36,37], produced during muscle cooking and headspace extraction. The mean amount of these compounds tended to be lower in the GS-LC meat and the sum of aldehydes was significantly reduced in this group. The greater intake of silage in the animals in this group would result in reduced thermal oxidation of lipids in beef. Despite the existing differences in the FA percentages between the experimental feeding systems, in contrast with the findings by Elmore et al. [38] and Mezgebo et al. [39], lipid-derived volatile compound concentrations were not related to the percentages of the major PUFAs, i.e., 18:2n − 6 and 18:3n − 3, in beef nor to the intramuscular fat content. This might be more related to intramuscular fat content. Finally, the scarce significant differences did not allow us to hypothesise that beef odour could be significantly changed by the feeding treatments.

### 3.2. Perirenal Fat Characteristics and Differentiation between Feeding Systems

The main aim of the analysis of perirenal fat was to evaluate the potential of the compositional and spectral characteristics of this fat depot, which is easy to sample in the slaughterhouse, with little damage to the carcass value, for differentiating between the feeding regimens, i.e., the amount of grass silage in the diet. As a first approach, from a previous study, the restriction of concentrate in the silage-fed animals resulted in a reduced fat content in perirenal fat, as well as perirenal fat percentage in the carcass [18].

Regarding the FA profile (Table 4), larger differences between feeding regimes for the major FAs were found between the grass silage-fed animals, i.e., GS-AC and GS-LC, and the straw group (Str-AC). Grass silage feeding (GS-AC and GS-LC) showed significantly greater levels in beef of SFA (*p* = 0.04 and *p* = 0.04), iso-14 (*p* = 0.04 and *p* = 0.003), 15 (*p* = 0.01 and *p* < 0.001) and 16 (*p* = 0.01 and *p* = 0.002), 17:0 (*p* < 0.001 and *p* = 0.03), 18:0 (*p* < 0.001 and *p* = 0.008), 20:0 (*p* = 0.04 and *p* = 0.007) and *c11*-18:1 (*p* = 0.004 and *p* = 0.04), and reduced levels of MUFA (*p* = 0.02 and *p* = 0.002), *t10*-,*t11*-18:1 (*p* < 0.001), PUFA (*p* < 0.001 and *p* = 0.002) and 18:2n − 6 (*p* < 0.001) than Str-AC. Comparing each of the grass silage-fed groups, the concentrate restriction (GS-LC), and thus an increased intake of silage, resulted in a greater level of 18:3n − 3 FA and a reduction of 18:1n − 9 (*p* = 0.01). The GS-LC group showed the highest levels of iso-17:0 and n − 3 FA content among the three feeding treatments.

The FA contents in this fat depot clearly showed that grass silage intake increased the levels of SFA, decreased the levels of PUFA, and increased those of n − 3 FA, as stated in other studies in beef comparing diets with increased levels of grass or grass silage [7,12,32,33]. However, as observed in intramuscular fat, the biohydrogenation intermediates were not positively related to grass silage feeding, i.e., the *t*-18:1 percentage was the highest in Str-AC, and the CLA percentage was not increased with greater silage intake, as it was not significantly different between GS-AC and GS-LC (*p* = 0.71 and *p* = 0.51 respectively).

Table 4 also shows the concentrations of β-carotene and retinol in perirenal fat. The β-carotene content differed between feeding regimes (*p* < 0.001). The content was much greater in the grass silage samples than in the straw samples. The amount of this compound in the GS-LC samples was more than twice that of GS-AC. In contrast, the amount of retinol did not show significant differences. The ranges of β-carotene content (µg/g perirenal fat) in the GS-AC, GS-LC, and Str-AC groups (not shown in tables) were, respectively, 0.11–0.39, 0.35–1.16, and 0.00–0.08. Only one GS-LC sample was inside the GS-AC range, and all the Str-AC samples were far below the ranges of the other two groups. Thus, β-carotene seems to be an outstanding compound for differentiating between silage-fed and straw-fed and is good for differentiating within the silage groups.

FA analysis of beef (perirenal fat) showed high differentiation power regarding the dietary background of cattle, i.e., beef from concentrate- and pasture-based feeding regimes, or grass-, partially grass- and concentrate-feeding regimes [10,13,14]. In the present study, a principal component (PC) analysis carried out using the percentages of the major FAs of perirenal fat (those with a percentage > 0.4 on total FAs) as variables (Figure 1) showed a neat separation between both silage groups (GS) and the straw group (Str-AC). Thus, the model was able to separate grass-silage-fed and non-grass-silage-fed animals.

The first PC, explaining 80.4% of the variance, was the main PC responsible for this separation. The FAs with the highest coefficients for the first PC were 18:0 and *t10*-,t11-18:1 (0.80 and −0.55, respectively). The second PC explained 13.7% of the variance and was mainly influenced by c9-18:1 (0.85) and *t10*-,*t11*-18:1 (−0.41). In agreement with Cama-Moncunill et al. [14], *t11*-18:1, in this case, the sum of *t11*- and *t10*-18:1 (both coeluting in the chromatogram), was an important predictor. However, in this study, the importance of 18:2n − 6 and 18:3n − 3 was overcome by 18:0 and c9-18:1. Considering these results, two powerful potential indexes provided by the FA analysis for differentiation between the silage- vs straw-feeding regimes would probably be the ratios 18:0/*t10*-,*t11*-18:1 and *c9*-18:1/*t10*-,*t11*-18:1. The ranges for the 18:0/*t10*-,*t11*-18:1 ratio were 5.0–7.6, 4.4–6.5, and 2.2–4.9 for GS-AC, GS-LC, and Str-AC, respectively, and those for *c9*-18:1/*t10*-,*t11*-18:1 were 4.3–7.0, 3.4–4.8, and 2.1–2.9, respectively (not shown in tables).

The number of volatile compounds found in the headspace of perirenal fat was 39 in total: 13 aldehydes, 3 alcohols, 12 aliphatic hydrocarbons, 7 FAs and lactones, 2 benzene compounds, and 2 terpene compounds. Table 5 shows the compounds with a concentration in the headspace higher than 0.5 ng equivalents of undecane/mL. Although the mean values of most of the compounds tended to be lower in the GS-LC group (as previously seen for the volatile muscle compounds), no significant differences were found. Thus, the volatile composition would not be useful for feeding system differentiating purposes.

Spectroscopy methods were applied to analyse the perirenal fat samples from the feeding regimes evaluated. The L*, a*, and b* values, as well as the I450-510 carotenoid pigment index [3], all extracted from the VIS spectra, are shown in Table 6. As shown, a*, b*, and I450-510 were significantly greater in the concentrate group (GS-LS) compared with the other two groups. These three values were correlated among them and b* and I450-510 with the β-carotene content. The Pearsons’ r values for the comparisons between b* and a*, b* and I450-510, b* and β-carotene content, and I450-510 and β-carotene content were 0.88 (*p* < 0.001), 0.79 (*p* < 0.001), 0.83 (*p* < 0.001), and 0.75 (*p* < 0.001), respectively. There was no overlap of samples between GS-LC and Str-AC ranges for b* and I450-510 values; for GS-LC and Str-AC, b* ranged from 10.2–13.7 and 6.8–10.0, respectively, and I450-510 ranged from 186–257 and 95–167, respectively. However, a considerable portion of the GS-AC samples overlapped within the ranges of the other groups.

VIS and NIR spectra were obtained from each fat sample. Figure 2A shows the average VIS spectra (360–740 nanometers), and Figure 2B shows the average NIR spectra (1100–2500 nanometers) for the three sample groups. PC analyses were performed with the VIS and NIR spectral data individually and considering both spectra jointly (VIS + NIR). PC analyses were also carried out from the first derivatives of absorbance with respect to wavelength. In any of the PC plots obtained, a neat separation of samples was observed according to the feeding regimes.

Only the distribution of samples on the plane formed by the two PCs of the VIS + NIR spectra and its first derivatives are shown in Figure 3A,B for brevity; these plots were the best in differentiating performance. In VIS + NIR, the two first PCs explained 96.6% of the variability. Most of the Str-AC samples were separated from those of GS-LC (Str-AC points towards the upper–right region and GS-LC at a lower position). In the VIS + NIR derivative, the PCs explained 93.4% of the variation and Str-AC tended to be above the 0-scale value of PC2, and most of GS-LC were below this value. In both plots, the GS-AC samples tended to be intermediate between the samples from the other two groups.

The NIR technique applied to perirenal fat depots has proved to be useful for authenticating the dietary background of ruminants [10], and for estimating the proportions of PUFA in beef [40]. In the same way, Dian et al. [41] found in the VIS spectra in the range of 400–700 nm and the combined VIS + NIR spectra in the range (400–2500 nm) used on perirenal fat successful to differentiate between meat from pasture-fed and concentrate-fed lambs, i.e., a correct classification of samples greater than 90%. In the present study, in contrast, neither VIS nor NIR or VIS + NIR spectra data seemed very useful for differentiation between the experimental feeding regimes.

Two possible explanations may account for the poor performance obtained. One might be that the differences in FAs and β-carotene in the beef between the experimental groups were not high or neat enough to produce significant spectral variation. In addition, active functional groups in different molecules showing variations between samples, e.g., water, different FAs, haeminic pigments, and collagen, could have interfered at different regions of the spectra [41], thus diminishing the technique’s differentiation power.

## 4. Conclusions

The effect of dietary grass silage level on the FA composition of beef was stronger and neater in the perirenal fat than in the intramuscular fat. Grass silage plus concentrate feeding as compared to straw plus concentrate feeding decreased the n − 6 FA content and n − 6/n − 3 ratio in the intramuscular fat, while in perirenal fat increased SFA and decreased MUFA, PUFA, and n − 6 FA contents and the n − 6/n − 3 ratio. Moreover, in both intramuscular fat and perirenal fat silage grass in the diet showed a decreasing effect on the levels of biohydrogenation intermediates *t10*- + *t11*-18:1. To increase the grass silage intake by restricting the concentrate offered to the young bulls did not practically affect the FA composition neither in intramuscular fat not in perirenal fat. Beef from grass silage-fed bulls and straw-fed bulls can be differentiated by PC analysis of FA percentages in perirenal fat, being the most important predictors of the percentages of 18:0, *t*-18:1, and *c9*-18:1. Moreover, β-carotene content, b* colour value and carotenoid colour index in perirenal fat can also be useful for grass silage-fed vs. straw-fed differentiation purposes. However, the VIS or NIR spectra data showed poor differentiating performance between grass silage-fed vs straw-fed diets, and volatile composition did not have appreciable differentiating power. The performance of the above-mentioned analysis in perirenal fat to differentiate between the two grass silage diets (with different grass silage levels) was, in general, not satisfactory, having the best for b* and the carotenoid colour index.

## Figures and Tables

**Figure 1 animals-13-00261-f001:**
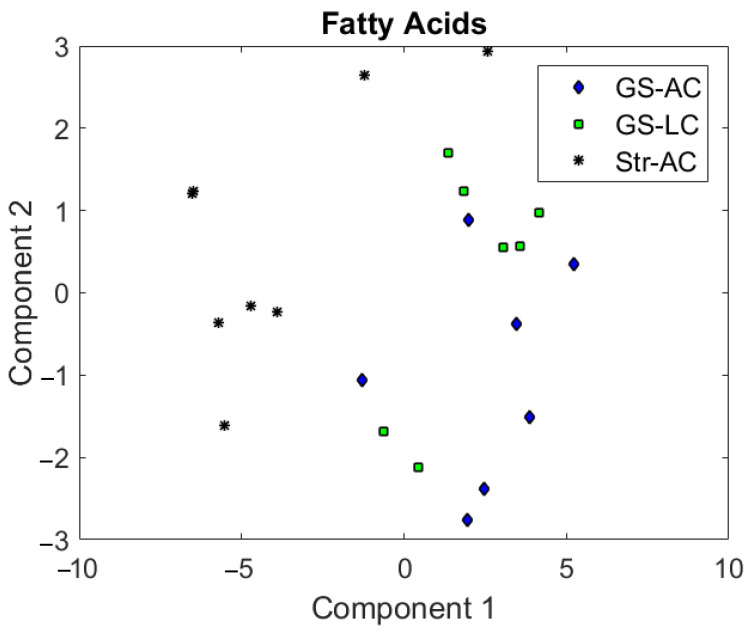
Principal component score plot using the main fatty acids (>0.4% on total fatty acids) as variables. GS-AC = grass silage plus concentrate ad libitum; GS-LC = grass silage plus restricted concentrate; Str-AC = barley straw plus concentrate ad libitum.

**Figure 2 animals-13-00261-f002:**
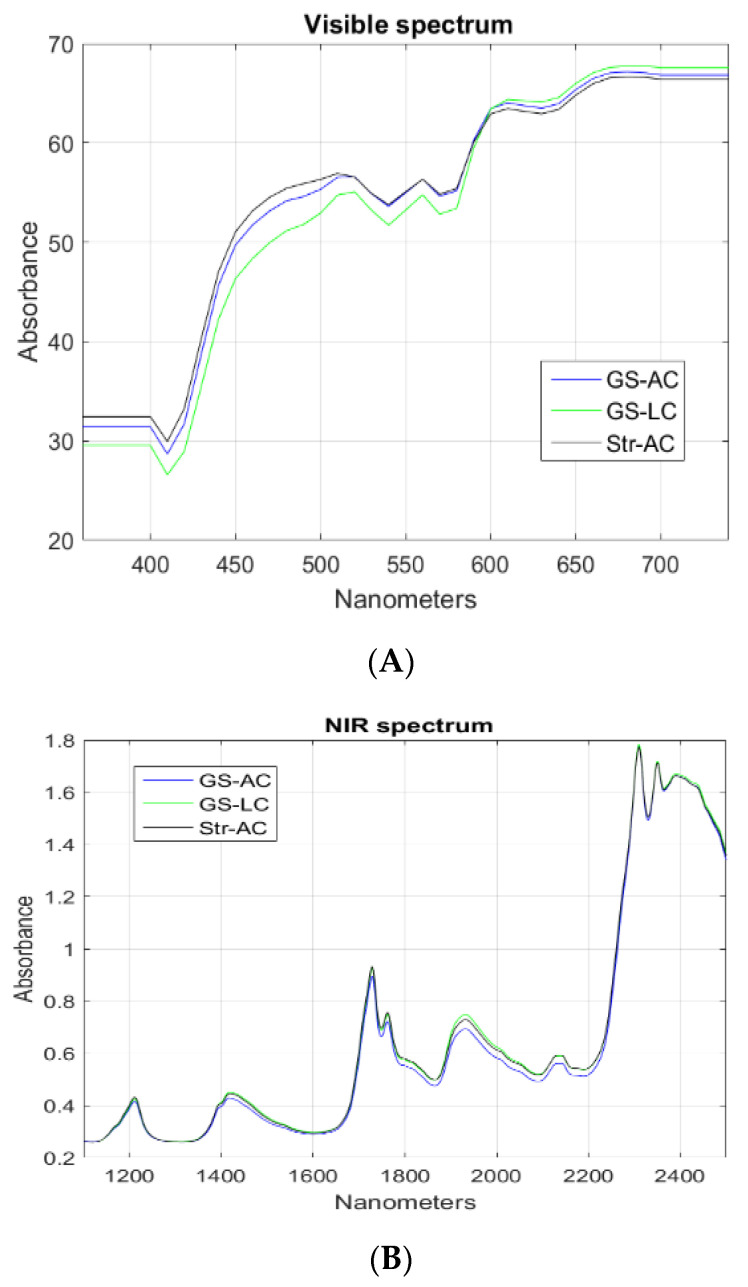
(**A**,**B**). Average VIS and NIR spectra. GS-AC = ad libitum grass silage plus ad libitum concentrate diet; GS-LC = ad libitum grass silage plus limited concentrate diet; Str-AC = barley straw plus ad libitum concentrate diet.

**Figure 3 animals-13-00261-f003:**
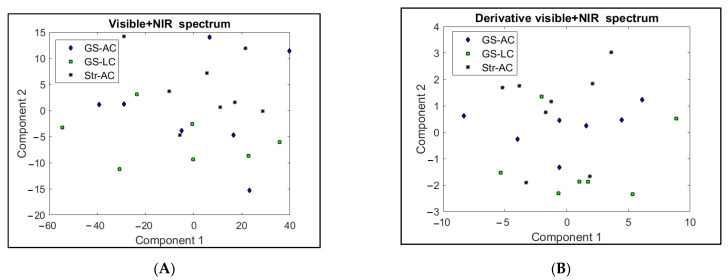
(**A**,**B**). Principal component score plots using data from the VIS + NIR spectra and its derivative as variables. GS-AC = ad libitum grass silage plus ad libitum concentrate diet; GS-LC = ad libitum grass silage plus limited concentrate diet; Str-AC = barley straw plus ad libitum concentrate diet.

**Table 1 animals-13-00261-t001:** Proximate composition (percentage of dry matter) ^&^, major fatty acids (percentage of total fatty acids) ^&^, and fat-soluble vitamins concentrations (µg/g of dry matter) of the animal feeds used for fattening the young bulls.

	Concentrate (*n* = 2)	Silage (*n* = 2)	Straw (*n* = 2)
Dry matter (DM)	90.4 ± 2.3	27.1 ± 2.9	88.7 ± 4.1
Crude protein	14.5 ± 0.4	10.1 ± 3.0	4.4 ± 0.5
Ether extract	3.7 ± 0.3	ND	ND
Acid detergent fibre	13.0 ± 1.8	32.4 ± 3.8	49.8 ± 3.9
Neutral detergent fibre	24.1 ± 2.9	59.2 ± 2.5	80.8 ± 6.2
Ash	6.2 ± 0.8	11.0 ± 2.6	7.6 ± 1.4
Net energy (milk feed units/ kg DM)	0.99 ± 0.04	0.75 ± 0.05	0.43 ± 0.04
14:0 ^&^	ND	2.5 ± 0.5	ND
16:0 ^&^	17.7 ± 1.7	23.9 ± 3.0	25.7 ± 2.1
18:0 ^&^	3.4 ± 0.3	3.2 ± 0.6	8.2 ± 0.5
*c9*-18:1 ^&^	20.1 ± 2.3	5.7 ± 0.9	13.8 ± 1.5
18:2n − 6 ^&^	55.9 ± 4.0	14.8 ± 1.9	23.5 ± 2.0
18:3n − 3 ^&^	2.9 ± 0.5	49.9 ± 9.8	28.8 ± 3.8
β-carotene	ND	79.9 ± 9.2	ND
Retinol	2.4 ± 0.8	ND	ND
α-tocopherol	40.0 ± 3.8	37.0 ± 9.0	1.9 ± 0.3
γ-tocopherol	2.2 ± 0.5	1.3 ± 1.0	1.1 ± 0.6
δ-tocopherol	0.22 ± 0.05	0.11 ± 0.04	ND

ND: not detected; ^&^ Values obtained from [18].

**Table 2 animals-13-00261-t002:** Fatty acids (FA), fat-soluble vitamins and oxidation stability in *longissimus thoracis* muscle from young Tudanca bulls reared on different feeding systems.

	GS-AC (*n* = 7)	GS-LC (*n* = 7)	Str-AC (*n* = 8)	SEM	*p*-Value
Fatty acids (mg/100 beef)					
SFA	1787.0	1103.5	1310.0	0.637	0.12
BCFA	3.2	3.0	4.7	0.020	0.69
OCFA	83.8	75.6	103.5	0.150	0.40
12:0	ND	ND	1.0	0.007	-
14:0	87.6	42.2	51.2	0.127	0.08
iso-C15:0	3.2	3.0	4.7	0.020	0.69
15:0	46.0	45.1	53.0	0.113	0.64
16:0	962.5	576.2	671.0	0.551	0.11
17:0	19.7	14.3	24.8	0.053	0.32
18:0	668.5	422.1	504.5	0.288	0.11
20:0	3.0	1.1	2.9	0.012	0.18
22:0	ND	2.5	1.6	0.025	-
MUFA	1602.2	982.1	1255.7	0.457	0.14
*c9*-14:1	8.0	3.0	4.4	0.028	0.23
*c9*-16:1	105.3	56.2	69.5	0.105	0.12
*c9*-17:1	15.0	13.2	21.0	0.048	0.43
*t10-*,*t11*-18:1	58.9 ^b^	50.8 ^b^	133.1 ^a^	0.322	0.02
*c9*-18:1	1374.6	821.4	949.2	0.619	0.08
*c11*-18:1	31.7 ^b^	34.2 ^b^	69.7 ^a^	0.186	0.04
*c12*-18:1	1.8	0.8	3.28	0.014	0.23
*c15*-18:1	2.9	0.2	1.9	0.016	0.17
*c11*-20:1	3.8	1.9	3.4	0.012	0.34
PUFA	416.8	383.0	566.5	0.877	0.06
n − 6	355.8 ^b^	327.7 ^b^	508.2 ^a^	0.816	0.03
n − 3	43.5	43.0	39.2	0.089	0.84
n − 6/n − 3	8.5 ^b^	7.6 ^b^	13.3 ^a^	0.622	<0.001
*t,t*-18:2 ^&^	3.1	2.2	3.1	0.020	0.49
18:2n − 6	272.4 ^b^	243.0 ^b^	386.4 ^a^	0.596	0.02
18:3n − 6	ND	0.01	0.04	0.006	-
18:3n − 3	20.6	18.8	15.3	0.040	0.37
*t,c*-*c,t*-18:2 ^&^	11.3	6.8	6.7	0.016	0.06
20:2n − 3	3.1	2.7 ^b^	6.0	0.022	0.055
20:2n − 6	1.4	3.1	2.5	0.020	0.055
20:3n − 6	13.5	13.0	20.6	0.043	0.057
20:4n − 6	65.8	67.2	90.9	0.177	0.10
20:5n − 3	6.5	8.9	8.5	0.030	0.34
22:4n − 6	4.1 ^b^	4.4 ^b^	9.2 ^a^	0.028	0.01
22:5n − 3	16.3	15.3	15.3	0.028	0.92
Vitamins (µg/g muscle)					
β-carotene	0.60	0.63	ND	0.077	-
Retinol	0.25	0.21	0.21	0.015	0.42
α-tocopherol	5.68	4.71	4.34	0.332	0.25
TBARS ^$^					
Day 0	1.10	0.95	1.37	0.070	0.08
ΔTBARS	0.40	0.48	0.44	0.077	0.77

GS-AC = grass silage plus concentrate ad libitum; GS-LC = grass silage plus restricted concentrate; Str-AC = barley straw plus concentrate ad libitum. SFA, BCFA, OCFA, MUFA, PUFA: saturated, branched-chain, odd-chain, monounsaturated, and polyunsaturated FA. SEM: standard error of the mean. ^&^ It may include several isomers. ^$^ Thiobarbituric reactive substances (TBARS) expressed as mg of malondialdehyde/kg beef on cooked beef just after cooking (Day 0) and the increment after two days of refrigerated storage (ΔTBARS). ND: not detected. ^a,b^: values in the same row with different letters are significantly different (*p* < 0.05) by the Fisher least square difference test.

**Table 3 animals-13-00261-t003:** Major headspace volatile compounds (ng equivalents of undecane/mL) in cooked longissimus thoracic muscle from young Tudanca bulls reared under different feeding systems.

	GS-AC	GS-LC	Str-AC	SEM	*p*-Level
Aldehydes	516.2 ^a^	241.3 ^b^	529.9 ^a^	50.23	0.046
Pentanal (701) ^&^	52.9	18.6	49.9	6.95	0.08
Hexanal (803)	341.2	152.7	327.4	36.26	0.08
Heptanal (901)	57.9	21.7	59.8	7.58	0.06
2-Heptenal (958)	0.7	0.1	0.7	0.14	0.11
Octanal (1002)	28.6	14.2	40.0	4.76	0.08
Octenal (1059)	1.2	0.3	1.3	0.22	0.12
Nonanal (1103)	30.1	27.7	39.0	3.75	0.38
2-Nonenal (1160)	0.4	0.1	0.5	0.09	0.08
Decanal (1204)	1.1	1.6	1.6	0.19	0.40
Dodecanal (1406)	0.2	0.5	0.4	0.09	0.49
Alcohols	23.6	17.2	28.0	3.17	0.39
Pentanol (775)	9.3	2.9	7.6	1.32	0.13
Hexanol (875)	1.3	0.7	1.6	0.24	0.26
Heptanol (973)	1.9	1.7	2.3	0.55	0.90
1-Octen-3-ol (981)	6.9	4.6	8.2	1.06	0.39
2-Ethyl-hexanol (1029)	3.1	4.7	4.8	1.25	0.24
Octanol (1073)	1.8	2.4	3.3	0.43	0.36
Ketones	79.6	52.4	56.5	7.06	0.20
3-Hydroxy-2-butanone (740)	2.3	2.4	7.2	1.33	0.22
Heptanone (890)	2.2	1.7	3.0	0.32	0.25
Octanedione (984)	5.2	2.9	5.6	1.15	0.20
Hydrocarbons	8.0	10.1	5.4	1.06	0.20
Octane (800)	3.0	6.3	3.1	0.56	0.051
2,2,4,6,6-Pentamethylheptane (985)	2.0	1.6	0.1	0.63	0.40
Unknown (1033)	1.4	0.2	1.0	0.28	0.22
Furans	8.8	3.6	9.6	1.47	0.21
2-Penthylfuran (988)	8.8	3.6	9.6	1.47	0.21
Fatty acids	4.1	3.3	2.8	0.50	0.57
Hexanoic acid (1008)	2.7	0.6	1.6	0.43	0.15
Dodecanoic acid (1691)	0.4	1.0	0.4	0.18	0.26
Benzene compounds	45.8	45.8	48.8	3.36	0.92
Toluene (767)	5.0	3.6	4.1	0.51	0.56
Benzaldehyde (965)	40.5	40.4	43.9	3.20	0.88
Sulphur compounds	10.2	5.3	7.6	1.58	0.48
Dimethyl disulfide (742)	3.0	1.6	1.9	0.56	0.61
Dimethyl trisulfide (970)	5.4	0.8	2.3	1.14	0.29

GS-AC = grass silage plus concentrate ad libitum; GS-LC = grass silage plus restricted concentrate; Str-AC = barley straw plus concentrate ad libitum. SEM: Standard error of the mean. ^&^ The experimental relative retention time for each compound is shown between brackets. ^a,b^ Means in rows showing any common superscript were significantly different (Fisher least square difference test; *p* < 0.05).

**Table 4 animals-13-00261-t004:** Fat content, fatty acid (FA) profile, and fat-soluble vitamins in the perirenal fat depot of young Tudanca bulls reared under different feeding systems.

	GS-AC	GS-LC	Str-AC	SEM	*p*-Level
Fat (% of fresh weight)	89.5 ^a^	87.3 ^b^	88.3 ^ab^	0.294	0.03
Fatty acids (mg/100 fat depot)					
SFA	58,095 ^a^	56,541 ^a^	53,371 ^b^	718.1	0.01
BCFA	2183	2442	2080	89.9	0.25
OCFA	3775	4275	4270	118.0	0.14
10:0	50	49	56	1.8	0.21
12:0	54	58	65	2.8	0.30
iso-14:0	44 ^a^	54 ^a^	29 ^b^	3.6	0.009
14:0	2759	2629	2897	82.0	0.42
iso-15:0	176 ^b^	200 ^b^	120 ^a^	10.9	0.002
anteiso-15:0	202	241	234	10.16	0.26
15:0	331	366	411	16.2	0.12
iso-16:0	185 ^a^	208 ^a^	135 ^b^	10.3	0.004
16:0	21,084	20,248	20,565	293.5	0.54
iso-17:0	364 ^b^	517 ^a^	369 ^b^	19.8	<0.001
anteiso-17:0	1119	1131	1113	54.3	0.99
17:0	1218 ^b^	1370 ^b^	1603 ^a^	52.2	0.003
iso-18:0	88	91	80	4.4	0.56
18:0	29,949 ^a^	28,849 ^a^	25,293 ^b^	644.2	0.002
19:0	81	95	90	7.3	0.75
20:0	340 ^a^	364 ^a^	282 ^b^	13.1	0.02
22:0	44	68	30	6.8	0.07
MUFA	31,599 ^b^	30,568 ^b^	33,825 ^a^	460.0	0.005
*c9*-14:1	136	115	151	8.5	0.24
*t9*-16:1	164	181	175	9.0	0.76
*c9*-16:1	621	665	683	48.7	0.88
*c9*-17:1	248	282	296	15.4	0.45
*t10*-,*t11*-18:1	4832 ^b^	5473 ^b^	8862 ^a^	444.8	<0.001
*c9*-18:1	25,040 ^a^	23,298 ^b^	23,252 ^b^	296.3	0.013
*c11*-18:1	463 ^a^	394 ^a^	257 ^b^	31.3	0.013
*c7*-19:1	19	18	14	7.2	0.97
*c8*-20:1	30	35	29	4.1	0.80
*c11*-20:1	46	104	105	14.0	0.15
PUFA	4321 ^b^	4459 ^b^	5667 ^a^	189.3	<0.001
n − 6	3260 ^b^	3388 ^b^	4778 ^a^	196.3	<0.001
n − 6/n − 3	6.97 ^b^	6.63 ^b^	8.35 ^a^	0.522	<0.001
*t,t*-18:2 ^&^	136	122	195	9.7	0.71
18:2n − 6	3232 ^b^	3370 ^b^	4698 ^a^	190.8	<0.001
18:3n − 3	473 ^ab^	514 ^a^	432 ^b^	12.7	0.02
*t,c*+*c,t*-18:2 ^&^	442 ^a^	365 ^ab^	322 ^b^	21.1	0.04
20:2n − 6	48	59	85	7.1	0.07
20:4n − 6	24 ^ab^	14 ^b^	39 ^a^	4.3	0.047
22:2n − 6	6 ^b^	5 ^b^	15 ^a^	4	0.04
Vitamins (µg/g perirenal fat)					
β-carotene	0.24 ^b^	0.58 ^a^	0.03 ^c^	0.060	<0.001
Retinol	1.94	2.12	1.79	0.200	0.22

GS-AC = grass silage plus concentrate ad libitum; GS-LC = grass silage plus restricted concentrate; Str-AC = barley straw plus concentrate ad libitum. SFA, BCFA, OCFA, MUFA, PUFA: saturated, branched-chain, odd-chain, monounsaturated, and polyunsaturated FA. SEM: standard error of the mean. ND: not detected. ^&^ it may include several isomers. ^a–c^ values in the same row with different letters are significantly different (*p* < 0.05) by the Fisher least square difference test.

**Table 5 animals-13-00261-t005:** Major headspace volatile compounds (ng equivalents of undecane/mL) in the perirenal fat headspace of young Tudanca bulls reared under different feeding systems.

	GS-AC	GS-LC	Str-AC	SEM	*p*-Level
Aldehydes	97.0	43.1	120.2	75.96	0.14
2-Pentenal (710)	2.2	1.8	2.4	1.41	0.70
Hexanal (798)	71.7	32.4	93.1	58.65	0.13
2-Hexenal (842)	1.1	0.2	1.2	1.01	0.17
Heptanal (895)	16.4	5.1	14.5	10.81	0.11
2-Heptenal (950)	1.6	0.8	2.6	1.90	0.17
Octanal (998)	1.8	1.0	2.6	1.97	0.31
Nonanal (1100)	1.2	1.0	1.8	1.15	0.39
2-Nonenal (1157)	0.6	0.4	0.8	0.51	0.21
Alcohols	3.1	1.0	3.0	2.15	0.19
Pentanol (772)	2.6	1.0	2.4	1.74	0.15
Hydrocarbons	15.1	8.5	8.6	6.81	0.35
Octadiene (783)	0.8	0.3	0.4	0.55	0.18
Octene (791)	5.5	3.6	2.6	2.64	0.50
Nonene (885)	1.9	1.0	1.1	0.97	0.16
2,2,7,7-Tetramethyl-octane (982)	3.1	2.1	2.4	1.43	0.43
Decene (986)	1.0	0.8	0.7	0.61	0.57
Acids and lactones	4.2	4.4	6.3	2.98	0.36
Hexanoic acid (999)	2.9	3.2	4.9	2.47	0.27
Benzene compounds	1.9	2.9	1.0	2.86	0.58
Toluene (768)	1.6	2.3	0.6	2.87	0.51

RRI: relative retention index. GS-AC = grass silage plus concentrate ad libitum; GS-LC = grass silage plus restricted concentrate; Str-AC = barley straw plus concentrate ad libitum. SEM: standard error of the mean. The experimental relative retention time for each compound is shown between brackets.

**Table 6 animals-13-00261-t006:** Instrumental colour values and index in perirenal fat of young Tudanca bulls reared under different feeding systems.

	GS-AC	GS-LC	Str-AC	SEM	*p*-Level
Colour					
L*	79.93	79.10	80.12	0.741	0.71
a*	0.98 ^b^	1.84 ^a^	0.79 ^b^	0.270	0.03
b*	9.79 ^b^	11.91 ^a^	8.40 ^b^	0.510	<0.001
I_450–510_ ^&^	171.1 ^b^	237.4 ^a^	122.8 ^c^	11.81	<0.001

GS-AC = grass silage plus concentrate ad libitum; GS-LC = grass silage plus restricted concentrate; Str-AC = barley straw plus concentrate ad libitum. SEM: standard error of the mean. ^a,b,c^ means in rows showing any common superscript were significantly different (Tukey test; *p* < 0.05). ^&^ I450–510: index obtained from the colour spectra proposed by Prache et al. (2009) to quantify light absorption by carotenoid pigments.

## Data Availability

Not applicable.

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
