# Peer review of "Lipid Characteristics of the Muscle and Perirenal Fat in Young Tudanca Bulls Fed on Different Levels of Grass Silage"

_animals, 2023, doi:10.3390/ani13020261_

Round 1

Reviewer 1 Report

This is a very intresting study that is well written.

There are only a few comments

Line 102: Please define the dates that the feed samples were obtained. For a feeding study with such a long duration more samples should have been taken and a pooled samples analysed at the end.

Line 126: The sample was homogenised (minced) once as described in line 124. Did you minced it again?

Line 166...: Did you flatten the perirenal fat? Please provide some additional information on how you performed the measurement. This is not a measurement that is usually performed.

Statistical analyses

Please include the Pearson correlation that you performed.

Line 503: The number of samples you used for NIR spectroscopy is extremely low and from the same treatments. You need at least 50 - 80 samples to construct a calibration curve.

Conclusions

It is rather long. Please be more focused.

Other points

Please provide some information of the fatty acid composition of the actual diets.

What was the actual TBARS level of the cooked samples? Maybe you should consider providing this information. Was it below the threshold value that consumers perceive the rancid flavour?

Author Response

Many thanks for your comments. Please find below the corresponding point by point answers.

Line 102: Please define the dates that the feed samples were obtained. For a feeding study with such a long duration more samples should have been taken and a pooled samples analysed at the end.

Answer: the data were provided. We will consider to take feed samples more frequently for further studies.

Line 126: The sample was homogenised (minced) once as described in line 124. Did you minced it again?

Answer: We made a mistake and have corrected the sentence. Mincing was carried out only once.

Line 166...: Did you flatten the perirenal fat? Please provide some additional information on how you performed the measurement. This is not a measurement that is usually performed.

Answer: The perirenal fat depot was cut with a knive and the colour was determined on the cut surface. Further description of the procedure is given in the revised version.

Statistical analyses Please include the Pearson correlation that you performed.

Answer: Pearson correlation has now been included.

Line 503: The number of samples you used for NIR spectroscopy is extremely low and from the same treatments. You need at least 50 - 80 samples to construct a calibration curve.

Answer: I agree that the analysis should be more robust with a higher number of samples. However, we try to explore the differentiation power of the technique, i.e. if based on differences in the spectra the samples could be grouped with success by feeding treatment. Unfortunately, the method proved to be not able to differentiate and so it was not worthy for this purpose.

Conclusions It is rather long. Please be more focused.

Answer: Conclusions have been rewritten and now are more focused on the results.

Please provide some information of the fatty acid composition of the actual diets.

Answer: FA composition was provided in Table 1

What was the actual TBARS level of the cooked samples? Maybe you should consider providing this information. Was it below the threshold value that consumers perceive the rancid flavour?

Answer: the TBARS value for recently cooked meat has been provided. The literature reported that to give a threshold value is difficult and not accurate because it depends on several factors including methodology See Holman and Hopkins (2021) Meat Science 181, 108586 at https://www.sciencedirect.com/science/article/pii/S0309174021001625. Thus, we prefer not to give one.

Reviewer 2 Report

Overall, the article is well written and the conclusions match the data. Throughout, remove the use of higher or lower, these are ambiguous terms and have multiple interpretations. Please change to greater or reduced. The term "discriminating" is used throughout the manuscript, this is a confusing term and not sure what it means in regards to the article. I suggest changing to a different term that better represents the meaning you are trying to make. Bulls aren't typically used for meat consumption, so some justification on using bulls should be added.

Line 30 - how young? 

Line 45 - missing an "and" or "or" 

Line 55-57 - Several studies contradict your statement that FA composition is impacted, be careful about how much you state that grass is better than grain

Line 61-74 - this is just a list of studies using different technologies with no explanation to its importance to the study, remove or add further justification

Line 78-83 - Similar to the previous comment, need to state that the current study is in addition to source 18

Materials and Methods - animal information needs to be included, age, body weight, how long were they on trial diets, etc. Diet tables need to be included. 11 months is young for harvest, did this impact results? Table 2 is not necessary if you include this in the text. These diets are drastically different, is this a good comparison for the trial.

Line 139 - spell out FA

Results and Discussion - Please use actual p-values in the text and tables. Please use the same abbreviation for odd-chain FA, you have OFA and OFCA in some places. Weird symbol for alpha on carotene.

Lines 232-235 - Only a single sentence, combine with previous or next paragraph

Line 238-240 - This is just explains the different treatments.

Conclusions - Grass and grass silage are not the same thing. Make sure you reference grass silage when discussing your trial.

Author Response

Many thanks for your comments. Please find below the corresponding point by point answers.

Overall, the article is well written and the conclusions match the data. Throughout, remove the use of higher or lower, these are ambiguous terms and have multiple interpretations. Please change to greater or reduced. The term "discriminating" is used throughout the manuscript, this is a confusing term and not sure what it means in regards to the article. I suggest changing to a different term that better represents the meaning you are trying to make. Bulls aren't typically used for meat consumption, so some justification on using bulls should be added.

Answer: Higher and lower have been changed to greater or reduced. Discriminating has been replaced to differentiating. The bulls (non-castrated) were 11 years old (weaners or feeders). Now in the revised version the word weaners have been used in the text (abstract and materials and method).

Line 30 - how young? 

Answer: Provided in the text, 11 year old

Line 45 - missing an "and" or "or" 

Answer: The sentence has been changed for clarity.

Line 55-57 - Several studies contradict your statement that FA composition is impacted, be careful about how much you state that grass is better than grain

Answer: the word “positive” has been deleted from the sentence

Line 61-74 - this is just a list of studies using different technologies with no explanation to its importance to the study, remove or add further justification

Answer: a large part of the text has been deleted and the rest have been rewritten.

Line 78-83 - Similar to the previous comment, need to state that the current study is in addition to source 18

Answer: The text has been rewritten to clearly state that the current study is a continuation of the previous one.

Materials and Methods - animal information needs to be included, age, body weight, how long were they on trial diets, etc. Diet tables need to be included. 11 months is young for harvest, did this impact results? Table 2 is not necessary if you include this in the text. These diets are drastically different, is this a good comparison for the trial.

Answers:

The animal and diet information is provided; however briefly because this was provided in the previous study (Serrano et al. 2018; Ref. 18). The FA composition of the diet has been included in Table 1. For the proximal composition the reader is remitted to Serrano et al 2018). The mean daily intake of the feeds was also provided.

In Spain it is normal to harvest the weaners for beef (a reference has been added in the manuscript for supporting the statement). In the current study we wanted to know the effect of grass silage levels on the lipids and lipid derived compounds of beef, comparing a conventional feeding (straw), a feeding similar to the conventional but replacing the straw by silage, and a feeding with a greater intake of silage than that of the previous one.

Actually, the intake of silage per each treatment group was 0, 1.5 kg/day and 2.6 kg/day, respectively. When the experiment was planned we searched for applicability of the feeding systems in farms.

Table 2 has been deleted and the number of the following tables modified.

Line 139 - spell out FA

Answer: it was spelled in line 54

Results and Discussion - Please use actual p-values in the text and tables. Please use the same abbreviation for odd-chain FA, you have OFA and OFCA in some places. Weird symbol for alpha on carotene.

Answer: The p-values have been written in Tables and text and the other comments have been amended. Thank you.

Lines 232-235 - Only a single sentence, combine with previous or next paragraph

Answer. Done.

Line 238-240 - This is just explains the different treatments.

Answer: the sentence has been rewritten to make it less descriptive and provide discussion.

Conclusions - Grass and grass silage are not the same thing. Make sure you reference grass silage when discussing your trial.

Answer: We have revised all the manuscript (and specifically conclusions)  and made always reference that grass silage was used.

Reviewer 3 Report

The aim of the research was to investigate the effect of grass silage on beef quality, as well as the potential of the compositional and spectral characteristics of perirenal fat for discriminating between feeding systems. In my opinion, the conclusions of the conducted research are clear and result from the obtained research results. The material used for the research is sufficient, and the research methods have been selected appropriately. The arrangement of the figure and tables is clear and presents the obtained results very well. Discussing the results against the background of other authors is very detailed. The publications cited by the authors of the article are well-selected. For the most part, the authors refer to the latest knowledge published in renowned scientific journals.
As a remark, I suggest the authors verify the journal's template and adapt it to the formatting of the manuscript. For example, "p-value" should be reported in lowercase and italics, etc.

Author Response

Many thanks for your comment: As a remark, I suggest the authors verify the journal's template and adapt it to the formatting of the manuscript. For example, "p-value" should be reported in lowercase and italics, etc.

Answer: The manuscript has been revised to fit to the journal template. The format of p-value has been corrected. Extensive changes have been done regarding the format of references.

Round 2

Reviewer 2 Report

The manuscript is much improved.

Please include diet proximate composition in Table 1. Do not just reference the other paper. The table should indicate if the diets were isonitrogenous and isocaloric which would heavily impact the results and conclusions.

In the results section, use the actual p-values you present in the tables. Also, 2 decimal places is sufficient for actual p-values in the tables.

Author Response

Many thanks for your corrections and suggestions. These are the answer to the last round

Please include diet proximate composition in Table 1. Do not just reference the other paper. The table should indicate if the diets were isonitrogenous and isocaloric which would heavily impact the results and conclusions.

Answer: The proximate composition values are included

In the results section, use the actual p-values you present in the tables. Also, 2 decimal places is sufficient for actual p-values in the tables.

Answer: The actual p-values were written in the result sections and 2 decimals were used for p-values except when p<0.01, where 3 decimal positions were kept.